# Chemistry and Functions of Imported Fire Ant Venom

**DOI:** 10.3390/toxins15080489

**Published:** 2023-08-03

**Authors:** Jian Chen

**Affiliations:** Biological Control of Pests Research Unit, Agricultural Research Service, U.S. Department of Agriculture, Stoneville, MS 38776, USA; jian.chen@usda.gov

**Keywords:** venom alkaloids, venom proteins, predatory toxins, defensive compounds, chemical communication, external disinfectant, internal antibiotics

## Abstract

In the United States, imported fire ants are often referred to as red imported fire ants, *Solenopsis invicta* Buren, black imported fire ants, *S. richteri* Forel, and their hybrid (*S. invicta × S. richteri*). Due to their aggressive stings and toxic venom, imported fire ants pose a significant threat to public health, agriculture, and ecosystem health. However, venom plays a vital role in the survival of fire ants by serving various crucial functions in defense, foraging, and colony health maintenance. Numerous reviews and book chapters have been published on fire ant venom. Due to its medical importance and the expanding global distribution of these ants, fire ant venom research remains an active and highly productive area, leading to the discovery of new components and functions. This review summarizes the recent advances in our understanding of fire ant venom chemistry and its functions within fire ant colonies.

## 1. Introduction

Imported fire ants are significant pests due to their aggressive behavior and venomous stings. They pose a growing concern in many regions as they can have detrimental effects on human health, agriculture, and the environment. In the United States, imported fire ants are generally recognized as two species and their hybrid: red imported fire ants (*Solenopsis invicta* Buren), black imported fire ants (*S. richteri* Forel), and hybrid imported fire ants (*S. invicta* × *S. richteri*). These two fire ant species were introduced from South America into the southern United States in the early twentieth century [1]. The distribution of *S. invicta* significantly expanded to include states such as Alabama, Arkansas, California, Florida, Georgia, Louisiana, Mississippi, New Mexico, North Carolina, Oklahoma, South Carolina, Tennessee, Texas, Virginia, and Puerto Rico [2]. More recently, *S. invicta* was found in Kentucky [3], indicating its ongoing spread. On the other hand, *S. richteri* is limited to relatively small areas in northeastern Mississippi [4], northwestern Alabama [5], southwestern Tennessee [6], and southwest Kentucky [3]. Hybrid imported fire ants were first detected in 1985 [7] and are currently distributed in Alabama, Arkansas, Georgia, Mississippi, Tennessee, and Kentucky [3,6,8,9]. Due to their medical and economic significance and wider distribution, most current fire ant research focuses on *S. invicta*. Both species belong to the *Solenopsis saevissima* species group, and the taxonomy of this group has undergone numerous revisions. These two species were initially considered as two color variants in the same species *S. saevissima* and extensive interbreeding was reported between these two variants [10]. Buren separated them into two distinct species in 1972 [11]; however, the biochemical evidence for hybridization between these two species was not detected until 1985 [7].

Both species of imported fire ants and their hybrid exhibit social polymorphism, with the colonies having two social forms: monogyne and polygyne. Monogyne colonies have a single egg-laying queen, while polygyne colonies contain multiple queens that may not be related. Both monogyne and polygyne forms of *S. invicta* coexist in the USA and can be found in the same locations [12]. However, in the USA, *S. richteri* colonies are believed to be exclusively monogyne [13]. This suggests that either only monogyne *S. richteri* was introduced into the USA or both social forms were introduced, but only the monogyne form was successfully established.

Female fire ants, including workers (both normal workers and minims, the first batch of workers in an incipient colony), female alates, and queens, possess poison glands where the venom is produced. Although they may have species-, social form-, and caste-specific chemical profiles, fire ant venoms are characterized by a high proportion of water-insoluble alkaloids and a small number of water-soluble proteins. Extensive efforts have been made to identify the alkaloid components in both species and the hybrid. In fact, the identification of mixed hydrocarbon and alkaloid profiles in some fire ant samples led to the discovery of the hybrid imported fire ants in the United States [7]. New alkaloid components, such as piperidene and pyridine alkaloids, have been continuously discovered in fire ant venom [14,15,16,17,18]. Early research on venom proteins resulted in the identification of four allergenic proteins [19], and modern proteomic approaches provided a better understanding of the protein components in fire ant venom [20].

Venom serves various important functions in the biology of ants. Apart from its roles in capturing prey and defending colonies against intruders, competitors, and diseases, venom components can also play a role in chemical communication. Several fire ant pheromones have been associated with the poison gland [21,22,23,24,25]. Additionally, venom alkaloids act as kairomones, which are used by natural enemies such as phorid flies to locate ant hosts [26,27]. A recent discovery revealed that fire ants actually feed their nestmates with their own venom [28]. Numerous reviews and book chapters have been published on ant venom chemistry [29,30,31,32,33,34,35,36]. The objective of this review is to present the latest findings on new venom components in imported fire ants and their functional roles.

## 2. Venom Components

### 2.1. Venom Alkaloids

Extensive research has been conducted on the venom alkaloids of fire ants. Prior to the classification of red and black imported fire ants as distinct species, the first piperidine alkaloid, *trans*-2-methyl-6-n-undecylpiperidine, was identified in *Solenopsis saevissima* [37]. Subsequently, a series of 2-methyl-6-alkyl or alkenyl piperidine alkaloids were discovered in both *S. invicta* and *S. richteri* [32,38,39] (Table 1, Table 2 and Table 3, Figure 1). The alkyl or alkenyl side chains on position six of the piperidine ring can consist of nine, 11, 13, 15, or 17 carbons. To simplify the nomenclature, the carbon numbers are typically used to denote these alkaloids. For instance, C_13_ represents an alkaloid with a saturated 13-carbon side chain, while C_13:1_ represents an alkaloid with an unsaturated 13-carbon side chain containing one double bond. The absolute configuration of all the piperidine alkaloids in fire ants is consistently 2*R*,6*R* for *trans*-isomers and 2*R*,6*S* for *cis*-isomers [32]. These piperidine alkaloids were commonly named solenopsins, which were further categorized as solenopsin A, B, C, and D based on the length of the alkyl side chain on position six of the piperidine ring (A: C_11_, B: C_13_, C: C_15_, and D: C_17_).

Each species of fire ant has its own distinct piperidine alkaloid profile. In *S. invicta* workers, the dominant piperidine alkaloids are C_13_, C_13:1_, C_15_, and C_15:1_, while in *S. richteri* workers, the dominant alkaloids are C_11_, C_11:1_, C_13_, and C_13:1_. Interestingly, the piperidine alkaloid profile in the venom of hybrid imported fire ant workers appears to be a mixture resembling that of their parent species [7,40,41]. The double bonds on the side chains of the alkaloids are predominantly in the *cis-*configuration, but *trans*-isomers of 2-methyl-6-tridecenylpiperidine (C_13:1_) and 2-methyl-6-pentadecenylpiperidine (C_15:1_) have also been identified in the venom of *S. invicta* workers [16]. Both *cis*- and *trans*-isomers of piperidines are present, but *trans*-isomers are typically more abundant in workers. The major alkaloids found in the venom of female alates are *cis*- and *trans*-2-methyl-undecylpiperidine (*cis*- and *trans*-C_11_). The production of venom alkaloids in fire ants is influenced by age, body size, and season [42,43]. Workers of intermediate age produce more venom compared to old and young workers, and the ratio of saturated and unsaturated C_13_ and C_15_ alkaloids differs between minor and major workers [42]. In reproductive individuals, older alates have higher proportions of both *cis*- and *trans*-C_13_ piperidines than younger alates. After the mating flight, the newly mated queen exhibits a similar alkaloid profile to female alates, but the relative abundance of *trans*-2-methyl-undecylpiperidine (*trans*-C_11_) decreases as the colony develops (Figure 2). The venom alkaloid profile in *S. invicta* workers is also influenced by social form. Monogyne workers have higher C_13_:C_13:1_ and C_15_:C_15:1_ ratios compared to polygyne workers. However, polygyne workers have higher levels of unsaturated alkaloids regardless of the growth temperature, sampling seasons, or geographic location [44].

Almost 40 years after the identification of the first piperidine alkaloids, a series of related alkaloids called 2-methyl-6-alkyl (or alkenyl) piperidenes were characterized in the venom of both ant species and their hybrid [14,16,17,18] (Table 1, Table 2 and Table 3, Figure 1). These alkaloids include both Δ^1,6^ and Δ^1,2^ isomers of 2-methyl-6-alkyl or alkenyl piperidenes. Similar to piperidine alkaloids, the alkyl or alkenyl side chains on position six of the piperidene ring can have carbon chain lengths of 9, 11, 13, 15, and 17. Piperidene alkaloids are found in workers, female alates, and queens, but only 2-methyl-6-undecylpiperidene (both Δ^1,2^- and Δ^1,6^-C_11_) occurs in female alates and queens [45]. These alkaloids were identified much later than piperidine alkaloids likely due to their structural similarity, often co-eluting with piperidines on GC columns, and their lower abundance.

The nomenclature of piperidene alkaloids can be confusing. For instance, some compounds have been named piperideines, while others are called piperidienes. Different methods have been employed to specify the location of the double bond on the piperidine ring. For example, a compound such as 2-methyl-6-alkyl-6-piperidene may also be referred to as 2-methyl-6-alkyl-Δ^1,6^-piperidene, while 2-methyl-6-alkyl-1-piperidene could also be known as 2-methyl-6-alkyl-Δ^1,2^-piperidene. It is crucial to be aware of these different naming conventions when discussing and researching fire ant venom alkaloids and their related compounds. Clarity and consistency in the nomenclature will aid in the accurate understanding and communication of scientific information in this field.

The discovery of new alkaloid components in fire ant venom continues to this day. In 2019, pyridine alkaloids were detected in the venom of imported fire ants, including *S. invicta*, *S. richteri,* and the hybrid [15] (Table 1, Table 2 and Table 3, Figure 1). This discovery was made possible through the use of a unique technique called solid-phase microextraction (SPME) coupled with gas chromatography–mass spectrometry (GC–MS) and a modified thermal desorption process. The SPME fiber was loaded with venom secretion, and a series of consecutive GC–MS injections were performed, each with a partial desorption. This approach allowed for the identification of hidden pyridine alkaloid peaks that were previously masked by overlapping piperidine or piperidene alkaloid peaks. As a result, ten 2-methyl-6-alkyl (or alkenyl) pyridines were discovered for the first time in the venom of imported fire ants. This advancement in venom chemistry has provided valuable insights into the alkaloid composition of fire ant venom and expanded our understanding of its chemical complexity.

The role of minim workers in the establishment and development of a fire ant colony is important. However, the chemistry of their venom has received limited attention. Previous studies have reported that the major component in the venom of minim workers in *S. invicta* is the C_13:1_ piperidine alkaloid (Figure 2), and the presence of a piperidene alkaloid has been proposed [46]. However, there is currently no available information regarding the venom chemistry of minim workers in *S. richteri* and the hybrid imported fire ants. Furthermore, there is a lack of knowledge regarding the venom proteins in the venom of minim workers for both species and their hybrid. Further research is needed to explore the venom chemistry of minim workers in different fire ant species and gain a better understanding of their venom composition and potential functions.

Indeed, the biosynthesis of fire ant venom alkaloids remains a relatively unexplored area of research. While extensive studies have been conducted on the identification and characterization of venom alkaloids in fire ants, there is limited knowledge about the biosynthetic pathways responsible for their production. Only one publication focused on the biosynthesis of solenopsins in *S. geminata* has been reported thus far [47]. It was hypothesized that both *cis*- and *trans*-solenopsins are acetate derived, similar to other alkaloids found in insects, such as tetraponerine-8 and coccinelline. Solenopsins are biosynthesized first by the formation of long chain acid from the linear combination of acetate units, then followed by the loss of the carboxyl group, the introduction of an amino group, intramolecular cyclization, and a reduction in the imino group. This pathway was believed to be similar to the biosynthesis of the hemlock alkaloid coniine. Similar to how γ-coniceine serves as a precursor to coniine in hemlock, both Δ^1,2^ and Δ^1,6^ piperidenes are believed to serve as precursors to piperidine alkaloids in fire ants. The identification of 2-methyl-6-alkyl (or alkenyl) pyridine alkaloids in fire ants may add further complexity to the possible biosynthesis pathway of fire ant piperidine alkaloids, since the reduction from pyridine alkaloids can be another possible route for the biosynthesis of piperidine alkaloids.

Investigating the biosynthesis of venom alkaloids in fire ants could provide valuable insights into how these compounds are synthesized and regulated, shedding light on the mechanisms behind caste- and age-dependent profiles and social form-dependent variations in alkaloid composition. Understanding the biosynthetic pathways may also uncover new targets for intervention, offering potential new strategies for specific control measures in fire ants. Additionally, the genes involved in venom alkaloid biosynthesis could be heterologously expressed in microorganisms, enabling the production of alkaloids with a high yield and purity, which could facilitate further biological studies and introduce possibilities for their potential application in various domains, including the development of pesticides and antibiotics [33].

Since this review focuses on imported fire ants, other *Solenopsis* species were not included in this review. However, a Appendix A was added with references, which contains information about the venom alkaloid components in 17 other *Solenopsis* species. It is worth noting that in addition to piperidine alkaloids, other types of alkaloids occur in other *Solenopsis* ants, including indolizine, indolizidine, and pyrrolidine alkaloids.

### 2.2. Venom Proteins

Systemic allergic reactions to fire ant stings are observed in approximately 2% of victims [48]. Extensive research has been dedicated to identifying the allergenic components present in fire ant venom. Using techniques such as gel filtration and high-performance cation exchange chromatography, four worker allergens have been isolated and characterized: Sol i 1, Sol i 2, Sol i 3, and Sol i 4 [49]. The properties and details of these allergens have been extensively investigated and reviewed [36,50,51,52,53,54,55,56,57]. Among the worker allergens, Sol i 2 and Sol i 3 are the major proteins found in *S. invicta* venom, while Sol i 1 and Sol i 4 are present in smaller amounts [49]. Sol i 2 and Sol i 4 are considered to be among the most potent allergens [50]. It is worth noting that workers and queens exhibit different sequence isoforms for these venom proteins. For instance, the major protein in fire ant worker venom is referred to as Sol i 2w, while in fire ant queen venom, it is denoted as Sol i 2q. These isoforms show a sequence identity of approximately 75.6% [49]. Regarding Sol i 4, several minor isoforms have been identified, including Sol i 4, Sol i 4.01, Sol i 4.02, and Sol i 4 q (Sol i 4 from the queen) [51,58]. These isoforms of Sol i 4 have been the subject of rigorous research and investigation. Two additional isoforms of Sol i 2, namely Sol i 2X1 and Sol i 2X2, are listed in the National Center for Biotechnology Information (NCBI) sequence database. These isoforms were derived from the genome sequence of *S. invicta* and can be identified by their respective accession numbers, XP_011156049 and XP_011156057 [59]. The study of fire ant venom allergens is crucial for understanding the mechanisms underlying allergic reactions and developing diagnostic tools and potential therapies for individuals who are hypersensitive to fire ant stings.

Since milked venom or whole abdomens were used for studying the venom proteins [20,49,51,58], the glandular origin of these proteins has been questioned since fire ants release both the content of the poison sac and the Dufour’s gland through the sting apparatus. The poison gland origin of these proteins was recently confirmed using imaging mass spectrometry [59].

Our understanding of fire ant venom protein components has significantly improved through the extensive proteomic characterization conducted by dos Santos Pinto et al. [20] and Cai et al. [60]. In the dos Santos et al. study, 46 proteins were identified in the venom of *S. invicta*. These proteins included allergenic proteins, phospholipase A2, a growth factor, myotoxins, the phospholipase A2 inhibitor, thioredoxin peroxidase, neurotoxins, and the anemone cytolytic toxin. The most abundant proteins included a pseudechetoxin (PsTx)-like protein, a Scolopendra toxin-like protein, three different forms of venom Sol i 3, and venom Sol i 1. These 46 proteins were categorized into four groups: true venom components, housekeeping proteins, body muscle proteins, and proteins involved in chemical communication. The active but non-toxic venom components were further classified into three subgroups based on their potential functions: self-venom protection, colony asepsis, and chemical communication. The true toxins were classified into five other subgroups, including proteins influencing victim homeostasis, neurotoxins, proteins promoting venom diffusion, proteins causing tissue damage/inflammation, and allergens. In a recent study comparing the uniprot toxin database, Cai et al. [60] screened a total of 316 toxin-related unigenes and 47 proteins from a total of 33231 unigenes and 721 proteins and predicted the structure of calglandulin, venom Sol i 3, and the venom prothrombin activator hopsarin-D. They also found that *S. invicta* toxins contained phospholipase A2, one of the most prevalent proteins in bee toxins [61], which may have contributed to the cross-reactivity shown in the Sol i 1 protein of *S. invicta* and bees. They found calglandulin for the first time in *S. invicta* venom. This protein was associated with the secretion of toxins from the gland into the venom [62], indicating that calglandulin may play a role in the production of venom in *S. invicta*. In addition, a total of seven putative sequences in the transcriptome and three putative sequences in the proteome were identified as serine proteinase-like BMK-CBP in *S. invicta* venom, which has only been reported in Chinese red scorpion (*Buthus martensii* Karsch) venom [63]. They also investigated the structure of the venom prothrombin activator hopsarin-D, which served a similar function as mammalian coagulation Fxa [64].

The venom protein components in *S. richteri* can differ from those in *S. invicta*. While three homologous proteins were identified in *S. richteri* venom as Sol r 1, Sol r 2, and Sol r 3 [57], there is no equivalent of Sol i 4 in *S. richteri* venom. The hybrid imported fire ants possess unique venom proteins not found in their parent species. For instance, lateral flow immunoassays on the venom proteins revealed that hybrid imported ants from Tennessee contained Sol i 2, Sol r 2, as well as the proteins Solh2, Solh2Tr97, and Solr2A69 [65]. Solh2 and Solh2Tr97 were believed to be unique to the hybrid imported ants. Additionally, the venom proteins can have different sequence isoforms in different castes [49,51,58]. For example, Sol i 2 in worker venom and queen venom share only a 75.6% sequence identity, indicating potential caste-dependent venom functions in fire ants. Therefore, an extensive caste-differentiated proteomic characterization of venom in both species and their hybrid is necessary to fully comprehend the protein components in imported fire ant venoms.

### 2.3. Venom Peptides

With the significant advances in genomic, proteomics, and mass spectrometry techniques, numerous ant venom peptide toxins have been characterized [66,67,68,69,70,71]. However, the peptide components in fire ant venom have received minimum attention, which is likely due to the difficulty in obtaining a sufficient amount of fire ant venom that is free of alkaloids [20]. The existence of peptides in fire ant venom was clearly demonstrated in the first attempt of the proteomic characterization of fire ant venom. The MALDI-TOF MS spectrum of the whole fire ant venom showed a series of small proteins, or large peptides, which occurred at molecular weights smaller than 10 kDa [20]. More importantly, the presence of the atrial natriuretic peptide (ANP) was demonstrated in *S. invicta* venom, the first report of the ANP in Hymenoptera venom [20]. The ANP is a cardiac hormone that regulates the salt–water balance and blood pressure by stimulating renal salt, water excretion, and vasodilation [72].

## 3. Venom Functions

### 3.1. Predatory and Defensive Toxins

Animal venoms typically serve two primary functions: increasing their feeding efficiency and deterring potential enemies. The functions of ant venom alkaloids have been well summarized in a recent review on the biological diversity of ant alkaloids [34]. It is obvious that venom also plays an important role in chemical communication in ant societies. Ants can employ highly unique ways to utilize their venom. For example, venom can be used in social parasitism that occurs when an invading queen from one species (parasite) uses the workers of another species (host) to rear her reproductive offspring in the host nest. Venom can be used in every step of this social parasitism process, including host colony infiltration, usurpation, and integration [73,74,75,76,77,78].

Fire ants are omnivorous and have a diet that includes various animals [79]. While their strong mandibles may allow them to handle small prey without using venom, the ability to sting and inject venom enables them to prey on larger animals, including reptiles, birds, and even mammals. Venom serves as a predatory toxin, which may contribute to the success of fire ants as invasive species.

The defensive role of venom is equally crucial to the survival of fire ants. Fire ants face numerous natural enemies, even in territories where they are relatively new, such as the United States. Various animals prey on fire ants, including armadillos, antlions, spiders, birds, lizards, and dragonflies [80,81,82,83,84,85,86]. Additionally, other ants compete with fire ants for resources [87,88,89]. Infectious diseases pose a significant challenge to social insects in general, given their close social interactions and limited genetic diversity. Fire ants are no exception, especially since they construct mounds. The soil, being their natural habitat, harbors many entomopathogenic bacteria, fungi, and nematodes [90,91].

The injection of venom through worker stings appears to aid the colony in deterring intruders. However, the deterrent effect can also be achieved without physical contact by spraying the venom. In this case, fire ant workers exhibit a behavior known as “gaster flagging,” in which they raise and waggle their gasters with the stingers pointing upward. This behavior has also been observed in other ant species, such as *Monomorium minimum* [92]. A recent study has shown that the venom produced by new fire ant queens can also serve as a deterrent and assist her in establishing a new colony [93]. During this vulnerable stage in the life of a fire ant colony, the founding queen(s) face significant challenges [1]. After the female alates depart from the parent colonies in a mating flight and before the new queens have their first batch of worker adults, the newly mated queens must confront intense pressure from predators and competitors without the assistance of workers. Aggression from other ant species poses a major obstacle for the queen’s successful establishment of new colonies [94,95]. Interestingly, the venom of fire ant queens can incapacitate competing ants more rapidly than the venom of worker ants [93]. Synthetic venom alkaloids have been used to reproduce such effects, and solenopsin A (C_11_ piperidine alkaloid) appears to be an effective contact neurotoxin that potentially contributes to the successful establishment of a new fire ant colony.

To combat infectious diseases, ants commonly use gland secretions as external disinfectants [96,97,98,99]. The antimicrobial properties of fire ant venom alkaloids and their analogs have been extensively studied [100,101,102,103,104,105,106,107,108], including their activities against bacteria, fungi, microsporidia, and entomopathogenic pathogens. Recent research has provided evidence suggesting that fire ant venom may also function as an internal antibiotic [28]. Venom alkaloids have been found in the digestive systems of fire ants across all castes. Unlike the low quantities of venom observed on various surfaces, the concentration levels of venom alkaloids in the digestive system are higher and have been reported to be effective against various pathogens. Trophallaxis, the exchange of food or secretions among colony members, is involved in the transfer of venom alkaloids. Alkaloids have been discovered in the midguts of larvae, which do not produce alkaloids themselves. The workers donate their venom to female alates, which become the new queens after the mating flight. The venom alkaloids are then transferred to the larvae in the new colony through trophallaxis, indicating the occurrence of a transfer of worker alkaloids to a new colony. Once the first batch of worker adults (minims) emerge, they provide venom to the larvae in the colony. Over time, normal worker adults gradually replace the minims and become the primary venom donors in the colony. Considering the well-documented antimicrobial properties of venom alkaloids against various entomopathogens and their high concentration levels, it is likely that the venom in the digestive system of fire ants serves as an internal antibiotic. Residing in the soil may increase the chances of exposure to various pathogens for all the members of a fire ant colony, and having an antibiotic circulating in the digestive system of each individual at all times may provide an effective disease prevention mechanism at the colony level.

### 3.2. Chemical Communication

Pheromones are compounds/substances produced by organisms that are used to mediate intraspecific interactions. Multiple pheromones have been found in the venom of fire ant queens [21,22,23,24,25]. One example is the queen recognition pheromone, which elicits orientation and attraction in workers and is found in the poison sac of *S. invicta* queens [21,24,109,110,111] This pheromone also promotes the deposition of brood. It has been demonstrated that the alkaloids in the venom are not responsible for this pheromonal effect [21] (see Appendix A “for the chemical structures of the queen recognition pheromone components).

It is well known that the venom alkaloid profiles vary significantly between queens and workers, as well as between female alates and reproductive queens (Figure 2). It was observed that workers from two social forms exhibited distinguishable differences in their alkaloid profiles [44,112]. Workers from monogyne colonies had a higher proportion of saturated piperidines than workers from polygyne colonies, mainly due to differences in the saturated/total C_13_ piperidine ratios [112]. Both the C_13_:C_13:1_ and C_15_:C_15:1_ venom alkaloids ratios of monogyne workers were significantly higher than that of polygyne workers, but the sum of the proportions of unsaturated alkaloids in polygyne workers was significantly higher than that in monogyne workers, regardless of the growth temperature, sampling seasons, or geographic location [44]. Additionally, in polygyne colonies, it was found that the proportions of *cis*-piperidine alkaloids differed based on the Gp-9 genotype in non-reproductive queens, although this difference disappeared once they became functional reproducers. At 14 days old, Gp-9^BB^ queens have a higher proportion of *cis*-C_11_ than Gp-9^Bb^ queens. At this age, dealation can also change the proportion of *cis*-piperidines [112]. Considering that the genotype-specific piperidine differences reflect variations in the rates of reproductive maturation between queens, it was speculated that venom alkaloids may serve as signals of fertility for fire ant queens [112].

Venom alkaloids can be used by fire ant natural enemies as kairomones. Kairomones are defined as compounds emitted by an organism that are used to mediate interspecific interactions, benefiting the receiver while being harmful to the emitter. *Pseudacteon* phorid flies (Diptera: Phoridae), specific parasitoids in fire ants, were introduced into the United States from South America for fire ant biological control [113]. During an investigation into the mechanism used by phorid flies to locate their hosts, it was discovered that nine piperidine alkaloids (*cis*-C_11_, *trans*-C_11_, *cis*-C_13:1_, *cis*-C_13_, *trans*-C_13:1_, and *cis*-C_15_) and two piperidene alkaloids (2-methyl-6-n-pentadecenyl-Δ^1,6^-piperidene and 2-methyl-6-n-pentadecyl-Δ^1,6^-piperidene) elicited a significant electroantennogram (EAG) response in *P. tricuspis* [26]. This finding suggests that fire ant venom alkaloids may act as kairomones, attracting their natural enemies. During the EAG response test using extracts from different body parts, the authors divided the abdomen sample into three parts: a non-glandular part, a poison gland/sac, and a glandular part without the poison gland/sac. It was observed that extracts from the non-glandular part and the glandular part without the poison gland/sac also elicited an EAG response, although it was less intense compared to the extract from the poison gland/sac. The authors presented three possible explanations: (1) the compounds from other glands may also yield a positive EAG response, (2) the sting apparatus may contain venom alkaloids, and (3) alkaloid contamination from the poison gland/sac may occur during dissection. Considering the widespread presence of venom alkaloids in the digestive system [28], another plausible explanation is that both the non-glandular part and the glandular part without the poison gland/sac contain segments of the digestive system that contain venom alkaloids.

Interestingly, the study on the crystal structure of worker allergen, Sol i 2, revealed that it shares similarities with insect odorant-binding proteins (OBPs), leading to the hypothesis that Sol i 2 may have a role in capturing and/or transporting small hydrophobic ligands, such as pheromones [114]. It is noteworthy that venoms from worker ants and queens contain different isoforms, which supports this hypothesis since workers and queens may possess different pheromones in their venoms. Utilizing the known structure of the worker venom protein Sol i 2w, three-dimensional homology models were generated for the worker venom protein Sol i 4.02 as well as the two main venom proteins in queens and female alates, Sol i 2q and Sol i 2X1. Surprisingly, the models indicated that these proteins have relatively small internal hydrophobic-binding pockets that are obstructed by approximately ten amino acids in the C-terminal region. In order for these proteins to function as carriers of hydrophobic ligands, a conformational change would be necessary to displace the C-terminal region, similar to the mechanism observed in the silk moth pheromone-binding protein [59]. If the fire ant venom OBPs indeed serve as pheromone carriers in secretion, further characterization of these proteins could greatly contribute to the identification of *S. invicta* pheromones.

## 4. Conclusions

The complexity of the venom composition in ant colonies indicates its multifaceted functions. Understanding the components and functions of venom not only helps us grasp the sociobiology of ants but also presents new opportunities for utilizing ant venom components for human benefit. Identifying new components and investigating the functions of known components are equally essential. Fire ant venom holds practical applications in various fields. It can be utilized for developing natural pesticides, effective immunotherapy for individuals allergic to fire ant stings, and innovative antibiotics, including disinfectant products that prevent microbial biofilm formation. These potential applications have significant implications for pest control, healthcare, and hygiene. Moreover, mounting evidence suggests that venom plays a crucial role in social communication through pheromones. Exploring minor compounds becomes paramount for unraveling the complexities of social interactions within fire ant colonies, such as nestmate recognition, nuptial flight, and social immunity. However, characterizing compounds in low abundance, such as insect pheromones, poses substantial challenges, demanding an interdisciplinary approach integrating analytical chemistry, chemical ecology, neurophysiology, and genomics.

## Figures and Tables

**Figure 1 toxins-15-00489-f001:**
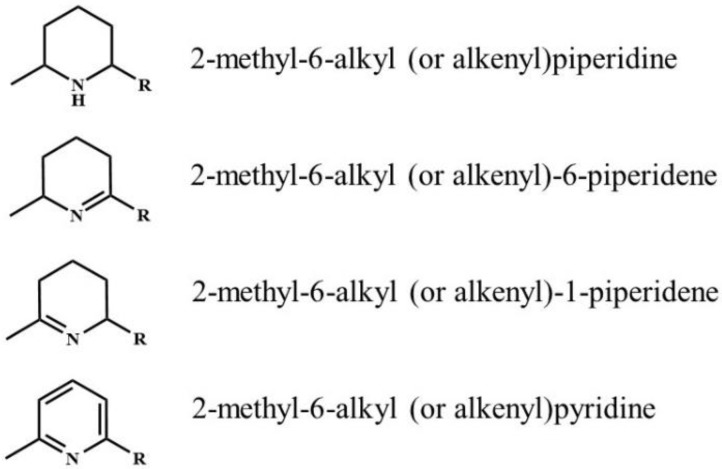
The general structures of fire ant venom alkaloids. R: the alkyl or alkenyl side chain on position six of the piperidine, piperidene, or pyridine ring, which can consist of 9, 11, 13, 15, or 17 carbons.

**Figure 2 toxins-15-00489-f002:**
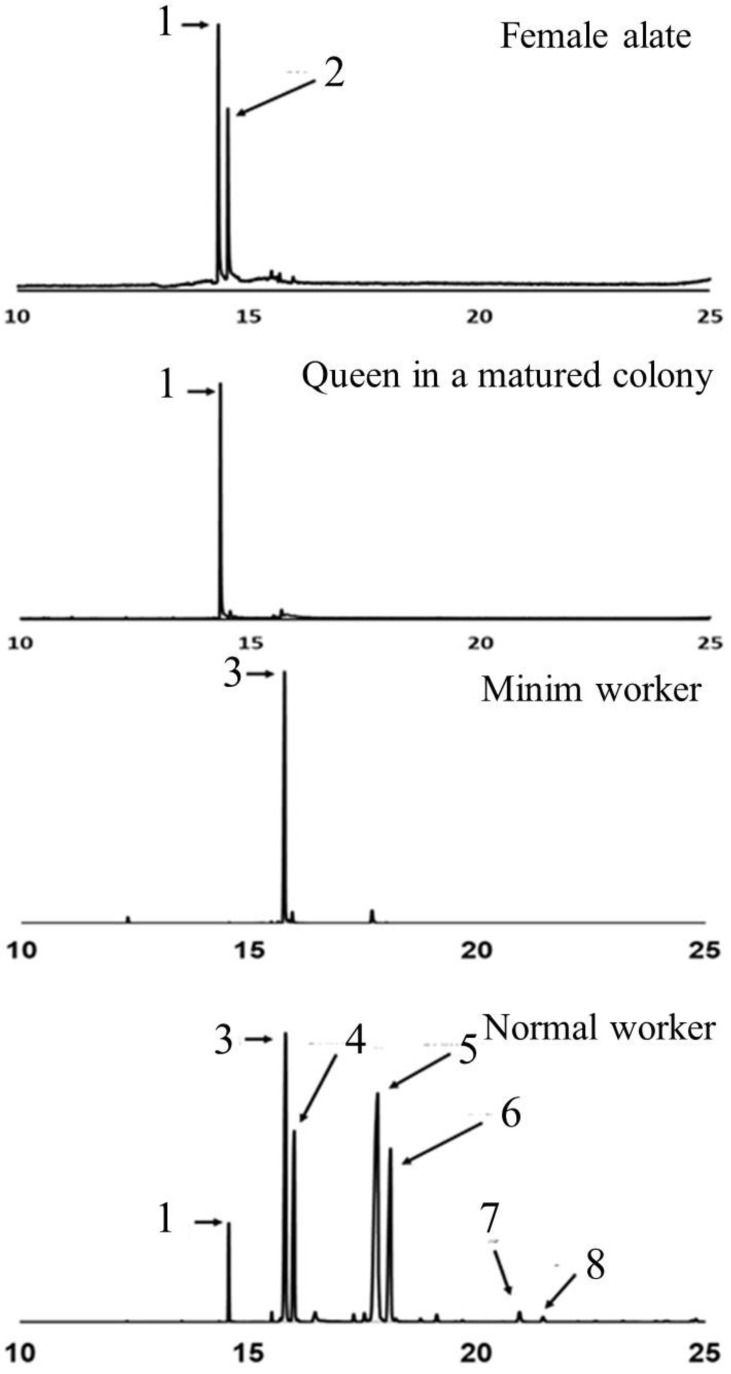
Chromatograms showing the major piperidine alkaloids in the venom of *Solenopsis invicta* female alates, the queen in a matured colony, the minim workers in the incipient colony, and normal workers in a matured colony, including 1: *cis*-C_11_, 2: *trans*-C_11_, 3: *trans*-C_13:1_, 4: *trans*-C_13_, 5: *trans*-C_15:1_, 6: *trans*-C_15_, 7: *trans*-C_17:1_, 8: *trans*-C_17_.

**Table 1 toxins-15-00489-t001:** Piperidine, piperidene, and pyridine alkaloids in *Solenopsis invicta*.

Alkaloid Type	Compound Name	*S. invicta*	
		Worker	Female Alate
Piperidine	*cis*-2-methyl-6-(undec-2-en-1-yl)piperidine	No	Yes
	*trans*-2-methyl-6-(undec-2-en-1-yl)piperidine	No	Yes
	*cis*-2-methyl-6-undecylpiperidine	Yes	Yes
	*trans*-2-methyl-6-undecylpiperidine	Yes	Yes
	*cis*-2-methyl-6-(tridec-3-en-1-yl)piperidine	No	Yes
	*cis*-2-methyl-6-(tridec-4-en-1-yl)piperidine	Yes	Yes
	*trans*-2-methyl-6-*(cis*-tridec-4-en-1-yl*)*piperidine	Yes	Yes
	*trans*-2-methyl-6-*(trans*-tridec-4-en-1-yl*)*piperidine	Yes	Yes
	*cis*-2-methyl-6-tridecylpiperidine	Yes	Yes
	*trans*-2-methyl-6-tridecylpiperidine	Yes	Yes
	*trans*-2-methyl-6-(tridec-8-en-1-yl)piperidine	Yes	No
	*cis*-2-methyl-6-(pentadec-6-en-1-yl)piperidine	Yes	Yes
	*trans*-2-methyl-6-*(cis*-pentadec-6-en-1-yl*)*piperidine	Yes	Yes
	*trans*-2-methyl-6-*(trans*-pentadec-6-en-1-yl*)*piperidine	Yes	Yes
	*trans*-2-methyl-6-*(*pentadec-14-en-1-yl*)*piperidine	Yes	No
	*cis*-2-methyl-6-pentadecylpiperidine	Yes	Yes
	*trans*-2-methyl-6-pentadecylpiperidine	Yes	Yes
	*cis*-2-methyl-6-(heptadec-8-en-1-yl)piperidine	Yes	No
	*trans*-2-methyl-6-*(*heptadec-8-en-1-yl*)*piperidine	Yes	No
	*cis*-2-methyl-6-(heptadecyl*)*piperidine	Yes	No
	*trans*-2-methyl-6-(heptadecyl*)*piperidine	Yes	No
Piperidene (Δ^1,6^)	2-methyl-6-undecyl-6-piperidene	Yes	Yes
	2-methyl-6-(tridec-4-en-1-yl)-6-piperidene	Yes	No
	2-methyl-6-tridecyl-6-piperidene	Yes	No
	2-methyl-6-(pentadec-6-en-1-yl)-6-piperidene	Yes	No
	2-methyl-6-pentadecyl-6-piperidene	Yes	No
	2-methyl-6-(heptadec-8-en-1-yl)-6-piperidene	Yes	No
	2-methyl-6-heptadecyl-6-piperidene	Yes	No
Piperidene (Δ^1,2^)	2-methyl-6-undecyl-1-piperidene	Yes	Yes
	2-methyl-6-(tridec-4-en-1-yl)-1-piperidene	Yes	No
	2-methyl-6-(tridecyl)-1-piperidene	Yes	No
	2-methyl-6-(pentadecyl)-1-piperidene	Yes	No
	2-methyl-6-(heptadecyl)-1-piperidene	Yes	No
Pyridine	2-methyl-6-tridecenylpyridine	Yes	Yes
	2-methyl-6-tridecenylpyridine-isomer 1	Yes	Yes
	2-methyl-6-tridecenylpyridine-isomer 4	Yes	Yes
	2-methyl-6-tridecylpyridine	Yes	Yes
	2-methyl-6-pentadecenylpyridine-isomer 2	Yes	No
	2-methyl-6-pentadecylpyridine	Yes	No

**Table 2 toxins-15-00489-t002:** Piperidine, piperidene, and pyridine alkaloids in *Solenopsis richteri*.

Alkaloid Type	Compound Name	*S. richteri*	
		Worker	Female Alate
Piperidine	*cis*-2-methyl-6-nonylpiperidine	Yes	Yes
	*trans*-2-methyl-6-nonyllpiperidine	Yes	Yes
	*cis*-2-methyl-6-(undec-2-en-1-yl)piperidine	No	Yes
	*trans*-2-methyl-6-(undec-2-en-1-yl)piperidine	Yes	No
	*cis*-2-methyl-6-(undecyl)piperidine	Yes	Yes
	*trans*-2-methyl-6-(undecyl)piperidine	Yes	Yes
	*cis*-2-methyl-6-(tridec-3-en-1-yl)piperidine	No	Yes
	*cis*-2-methyl-6-(tridec-4-en-1-yl)piperidine	Yes	Yes
	trans-2-methyl-6-(tridec-4-en-1-yl)piperidine	Yes	Yes
	*cis*-2-methyl-6-tridecylpiperidine	Yes	Yes
	*trans*-2-methyl-6-tridecylpiperidine	Yes	Yes
	*cis*-2-methyl-6-(pentadec-6-en-1-yl)piperidine	Yes	Yes
	*trans-*2-methyl-6-*(cis*-pentadec-6-en-1-yl)piperidine	Yes	Yes
	*cis*-2-methyl-6-pentadecylpiperidine	Yes	Yes
	*trans*-2-methyl-6-pentadecylpiperidine	Yes	Yes
Piperidene (Δ^1,6^)	2-methyl-6-undecyl-6-piperidene	Yes	Yes
	2-methyl-6-(tridec-4-en-1-yl)-6-piperidene	Yes	No
	2-methyl-6-tridecyl-6-piperidene	Yes	No
	2-methyl-6-(pentadec-6-en-1-yl)-6-piperidene	Yes	No
	2-methyl-6-pentadecyl-6-piperidene	Yes	No
Piperidene (Δ^1,2^)	2-methyl-6-undecyl-1-piperidene	Yes	Yes
	2-methyl-6-(tridec-4-en-1-yl)-1-piperidene	Yes	No
	2-methyl-6-(tridecyl)-1-piperidene	Yes	No
	2-methyl-6-(pentadec-6-en-1-yl)-1-piperidene	Yes	No
Pyridine	2-methyl-6-tridecenylpyridine	Yes	Yes
	2-methyl-6-tridecenylpyridine-isomer 1	Yes	Yes
	2-methyl-6-tridecenylpyridine-isomer 2	Yes	Yes
	2-methyl-6-tridecenylpyridine-isomer 3	Yes	Yes
	2-methyl-6-tridecenylpyridine-isomer 4	Yes	Yes
	2-methyl-6-tridecenylpyridine-isomer 5	Yes	Yes
	2-methyl-6-tridecylpyridine	Yes	Yes
	2-methyl-6-pentadecenylpyridine-isomer 1	Yes	No
	2-methyl-6-pentadecenylpyridine-isomer 2	Yes	No

**Table 3 toxins-15-00489-t003:** Piperidine, piperidene, and pyridine alkaloids in hybrid imported fire ants (*Solenopsis invicta* × *Solenopsis richteri*).

Alkaloid Type	Compound Name	*S. invicta* × *S. richteri*	
		Worker	Female Alate
Piperidine	*cis-*2-methyl-6-nonylpiperidine	No	Yes
	*trans*-2-methyl-6-nonyllpiperidine	No	Yes
	*cis*-2-methyl-6-(undec-2-en-1-yl)piperidine	No	Yes
	*trans*-2-methyl-6-(undec-2-en-1-yl)piperidine	Yes	No
	*cis*-2-methyl-6-(undecyl)piperidine	Yes	Yes
	*trans*-2-methyl-6-(undecyl)piperidine	Yes	Yes
	*cis*-2-methyl-6-(tridec-3-en-1-yl)piperidine	No	No
	cis-2-methyl-6-(tridec-4-en-1-yl)piperidine	Yes	Yes
	*trans*-2-methyl-6-(tridec-4-en-1-yl)piperidine	Yes	Yes
	*cis-*2-methyl-6-tridecylpiperidine	Yes	Yes
	*trans*-2-methyl-6-tridecylpiperidine	Yes	Yes
	*cis*-2-methyl-6-(pentadec-6-en-1-yl)piperidine	Yes	Yes
	*trans*-2-methyl-6-(pentadec-6-en-1-yl)piperidine	Yes	Yes
	*cis*-2-methyl-6-pentadecylpiperidine	Yes	Yes
	*trans*-2-methyl-6-pentadecylpiperidine	Yes	Yes
Piperidene (Δ^1,6^)	2-methyl-6-undecyl-6-piperidene	Yes	Yes
	2-methyl-6-(tridec-4-en-1-yl)-6-piperidene	Yes	No
	2-methyl-6-tridecyl-6-piperidene	Yes	No
	2-methyl-6-(pentadec-6-en-1-yl)-6-piperidene	Yes	No
	2-methyl-6-pentadecyl-6-piperidene	Yes	No
Piperidene (Δ^1,2^)	2-methyl-6-undecyl-1-piperidene	Yes	Yes
	2-methyl-6-(tridec-4-en-1-yl)-1-piperidene	Yes	No
	2-methyl-6-(tridecyl)-1-piperidene	Yes	No
	2-methyl-6-(pentadec-6-en-1-yl)-1-piperidene	Yes	No
	2-methyl-6-(pentadecyl)-1-piperidene	Yes	No
Pyridine	2-methyl-6-tridecenylpyridine	Yes	Yes
	2-methyl-6-tridecenylpyridine-isomer 1	Yes	Yes
	2-methyl-6-tridecenylpyridine-isomer 2	Yes	Yes
	2-methyl-6-tridecenylpyridine-isomer 3	Yes	Yes
	2-methyl-6-tridecenylpyridine-isomer 4	Yes	Yes
	2-methyl-6-tridecenylpyridine-isomer 5	Yes	Yes
	2-methyl-6-tridecylpyridine	Yes	Yes
	2-methyl-6-pentadecenylpyridine-isomer 1	Yes	No
	2-methyl-6-pentadecenylpyridine-isomer 2	Yes	No
	2-methyl-6-pentadecylpyridine	Yes	No

## Data Availability

Not applicable, since this is a review paper.

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
