# Peer review of "Chemistry and Functions of Imported Fire Ant Venom"

_toxins, 2023, doi:10.3390/toxins15080489_

Round 1

Reviewer 1 Report

This is a nice overview on fire ant venom chemistry and its functions within fire ant colonies. The structure of the manuscript is well organized and professionally presented. There are about 20 Solenopsis fire ant species or so. Less than half of these species are known for their venom chemistry. A whole picture of venom chemistry of all known species is still lacking. As several journal papers and book chapters on the imported fire ant chemistry have been published, it would attract more attentions if the venom chemistry of other fire ant species was included. The “venom functions” section is very simplified. It can be expanded.

Other specific comments.

Table 1: Would it be possible to provide structure of each component?

L81-83, L136-141: There were much more information available if other fire ant species were included.

L182: What does this Sol i 4q [45, 2012, 2018] mean?

L183: Sol i2

L201: What are venom allergen-3, and venom allergen-1? Are they Sol i3 or Sol i1?

Good.

Reviewer 2 Report

This detailed and clearly written review of the venom components found in imported fire ants will be of interest to fire ant venom researchers, and it provides a useful guide to help formulate new research questions. I have no comments and I couldn’t even find any language errors! Well done!

Author Response

Reviewer #2 does not have any comments.

Reviewer 3 Report

Dear authors and editor,

The receipt of this manuscript for review at this time comes as quite remarkable, given that I have just revised another manuscript for this very same journal addressing essentially the same topic. Please, consider the objective and scope of the just-published paper below, by Xu & Chen "Biological Activities and Ecological Significance of Fire Ant Venom Alkaloids" https://doi.org/10.3390/toxins15070439: "In this work, we review patterns of venom use, as well as alkaloids’ chemistry, toxicities, potential applications, and key ecological effect on fire ant communities".

Line 12 of the present manuscript states that "This review summarizes the recent advances in our understanding of fire ant venom chemistry and its functions within fire ant colonies."

Considering most of the fire ants venom is alkaloids, the two reviews cover much of the same ground. As a reviewer behind a double-blind interface, I cannot say whether the authors are also the same. The partial overlap being remarkable, I would ask the present authors to directly include a discussion on the paper by Xu & Chen (2023) and generally address their key contributions, by taking the opportunity to further expand and highlight their differential contribution in this submission relative to Xu & Chen's just-in publication on the alkaloids. Some of the mentioned topics in alkaloids are inevitably the same, such as the recent discovery of pyridines, the antimicrobial and deterrent functions, how alkaloids may attract phorid flies, etc.

This stated, the present manuscript is a well-written competent contribution on the topic fire ant venoms. In fact, this is a topic which has been growing in complexity and such reviews can attract more researchers and help readers keep updated with what is being done in the field. I will try to contribute with comments that might aid the authors to improve some specific aspects in their paper.

Please, upon starting on the topic of Venom Alkaloids, could the authors provide a formal definition for "solenopsins" in chemistry that would conform with the piperidines and recently discovered pyridines? Other organisms exist with alkaloids which are structurally close to solenopsins. This is a nomenclature uncertainty which has been growing that merits being addressed. 

Upon line 35-37, the authors make a point about how S. invicta was previously considered a red form of nominal species S. saevissima. This results in the fact that, on line 39, the authors state "... early literature on imported fire ants may 39 not provide clear information about the target ant species". This fact converges with the fact mentioned at line 74 that Solenopsin A was the 1st venom alkaloid, described from S. saevissima, from source reference no. 36. The authors are likely aware that S. saevissima (current sense) does not currently exist in Texas, US, therefore likely this was S. invicta.

 - On Tables 1 & 2, pages 85 & 88, please correct a typo on "Piperdine".

Please, could also the authors explain these differences between piperidines, piperideines and piperidenes, and add comments on some other compounds spelt "piperidienes" which have been only briefly mentioned in previous manuscripts, such as: Chen et al. (2019; 10.1021/acs.jafc.9b03631), Meer et al. (2022; 10.1007/s00114-022-01786-w) and Fox et al. (2019; 10.1016/j.toxicon.2018.11.428). The nomenclatures of these venom alkaloids is indeed confusing, and any systematic clarification would come as greatly useful.

In line 129 the authors mention piperideines took decades to get identified, however it should have been noted one of them was first described by Brand et al. 1972 in a different species of fire ant, who hinted they should be present also in imported species. Based on the title of one of their published papers, I suspect they were left without further notice simply because nobody revisited fire ant venom alkaloids IDs for decades after the detailed descriptions by Brand, Blum, Jones, MacConnell. Prof. Jones is still active and might be able to comment on his thoughts about why they were apparently overlooked. 

The section of venom proteins is quite well organized. There is, however, a major aspect that was not included from recent literature. The authors mention a proteomic study from 2012 in the paragraph 195-208, highlighting its contributions to the field. However, they overlooked an equally important paper by Cai et al. (2022; 10.1186/s12953-022-00197-z) also describing the presence of some 47 (other?) proteins. How did these results complement each other, and what insights can be made? Apparently neither Cai et al. (2022) discussed much about the interplay between these two papers on the same topic, thus it would be great if the present review offers some insights.

Notwithstanding: please mind to address what has been mentioned about fire ant peptides by previous authors after proteins.

On the section about the venom functions, authors make a point about fire ants behavior in the field and adaptations. However, there is the evident feeling that "fire ants" (in the scope of this review, meaning just a handful of Solenopsis species out of hundreds of similar related ants) are special or unique in their particulars. When actually they are trivially similar to other Solenopsidini in many aspects, including their venom use. This could be improved in some passages, such as the statement in line 240 that gaster-flagging would be a "unique behavior" of Solenopsis invicta. Gaster-flagging to spread alkaloids is not unique to any fire ant, for instance described by Adams & Traniello (1981; 10.1007/BF00540612) for Monomorium minimum. The deterring properties of alkaloid-loaded venoms of queens is immediately observed by anyone following a newly-mated mated thief-ant founder queen (see papers on other Solenopsidini like Megalomyrmex spp.). Therefore, if the authors could please better contextualise this section on venom ecology within what is generally known about related ants, and then possibly highlight what is really unique or remarkable about these (few) invasive fire ant species. Also the biosynthesis of the alkaloids is only superficially discussed where more could have been said about it, e.g. pointing out to structural similarities with coniine and other alkaloids where the pathways are better studied. How could these pathways relate with the piperideines and pyridines presented at length in the opening of this manuscript? A key reference on ant venom alkaloids is missing from the discussed literature, and contains useful references in all these aspects to be improved: Fox & Adams (2022; 10.1146/annurev-ento-072821-063525).

Finally, the section on chemical ecology could be significantly improved, mainly by adding further specific information on some important compounds mentioned. Please, could you provide more chemical information about the queen recognition pheromone (line 280), what was exactly the difference in alkaloids profile between different S. invicta genotypes, and Gp-9 (line 290)? These are interesting facts that were so superficially mentioned, constrasting with the kairomone function with phorid flies -- which was perhaps a bit too detailed on its part. Some more comments on the hypothetical role of OBPs in fire ant venom biology would also be of interest. 

In Conclusions, the authors again pushed the discovery of piperideines in fire ant venoms as remarkably novel, when Brand et al. (1972) had already pointed it out from another species and hypothesized it should be a precursor to piperidines present in fire ant venoms in general. Also, the report of piperideines in invasive fire ants from 2009 (coincidentially or not, also claimed concomitantly by two competing publications by different groups submitted almost simultaneously) no longer figures as "recent". In short, I really did not feel like the presented review evidentiated that investigations into the minor venom components have made sizeable breakthroughs. In fact, several of the most recent studies have described important functions in long-known compounds such as solenopsin A and Sol i 2. From experience, I would risk saying that major chemical aspects of these venoms are likely unreported, waiting on further creative methods of extraction & analysis. Therefore, I would suggest that the authors reconsider their conclusions to focus more on the bigger picture of the state of art of the literature. 

I am thankful for the opportunity to contribute, hopefully some suggestions prove useful to the authors and future readers.

Written English looks fine to me.

Round 2

Reviewer 3 Report

The authors have done a great job in preparing and revising this manuscript.  I am sure it will be useful to many readers in providing a comprehensive overview of a complex topic of interest.

I am thankful for the opportunity to contribute.